# Conditioned Medium of Human Pluripotent Stem Cell-Derived Neural Precursor Cells Exerts Neurorestorative Effects against Ischemic Stroke Model

**DOI:** 10.3390/ijms23147787

**Published:** 2022-07-14

**Authors:** Hye-Jin Hur, Ji Yong Lee, Do-Hun Kim, Myung Soo Cho, Sangsik Lee, Han-Soo Kim, Dong-Wook Kim

**Affiliations:** 1Department of Physiology, Yonsei University College of Medicine, Seoul 03722, Korea; bchhj@naver.com (H.-J.H.); scor1114@gmail.com (D.-H.K.); 2Brain Korea 21 Project for Medical Science, Yonsei University College of Medicine, Seoul 03722, Korea; 3Research Institute of Hyperbaric Medicine and Science, Yonsei University Wonju College of Medicine, Wonju-si 26426, Korea; hope9294@hanmail.net; 4S. Biomedics Co., Ltd., Seoul 04979, Korea; tpguy@sbiomedics.com; 5Department of Biomedical Engineering, College of Medical Convergence, Catholic Kwandong University, Gangneung-si 25601, Korea; lsskyj@cku.ac.kr; 6Department of Biomedical Sciences, College of Medical Convergence, Catholic Kwandong University, Gangneung-si 25601, Korea

**Keywords:** conditioned medium, neural progenitor cells, ischemic stroke, proteome analysis, transcriptome analysis, neurogenesis, inflammation

## Abstract

Previous studies have shown that early therapeutic events of neural precursor cells (NPCs) transplantation to animals with acute ischemic stroke readily protected neuronal cell damage and improved behavioral recovery through paracrine mechanisms. In this study, we tested the hypothesis that administration of conditioned medium from NPCs (NPC-CMs) could recapitulate the beneficial effects of cell transplantation. Rats with permanent middle cerebral artery occlusion (pMCAO) were randomly assigned to one of the following groups: PBS control, Vehicle (medium) controls, single (NPC-CM(S)) or multiple injections of NPC-CM(NPC-CM(M)) groups. A single intravenous injection of NPC-CM exhibited strong neuroregenerative potential to induce behavioral recovery, and multiple injections enhanced this activity further by suppressing inflammatory damage and inducing endogenous neurogenesis leading to histopathological and functional recovery. Proteome analysis of NPC-CM identified a number of proteins that are known to be associated with nervous system development, neurogenesis, and angiogenesis. In addition, transcriptome analysis revealed the importance of the inflammatory response during stroke recovery and some of the key hub genes in the interaction network were validated. Thus, our findings demonstrated that NPC-CM promoted functional recovery and reduced cerebral infarct and inflammation with enhanced endogenous neurogenesis, and the results highlighted the potency of NPC-CM in stroke therapy.

## 1. Introduction

Ischemic stroke, the most common cerebrovascular disease with high rates of mortality and long-term disability [1], is caused by abrupt blockade of cerebrovascular blood flow leading to rapid cell death and impairment of neural function in the affected lesion. Although a number of different therapeutic strategies have been proposed [2], intravenous injection of tissue plasminogen activator (tPA, alteplase) [3] is the only effective FDA-approved therapy for thrombolytic treatment of acute ischemic stroke. However, the therapeutic window of tPA is quite narrow since its clinical benefits manifest when the agent is administered within 4.5 h of stroke onset, and delayed tPA administration is associated with deleterious side effects [4,5]. Because of the absence of effective therapies that alleviate the long-term disability caused by acute or subacute ischemic stroke, effective therapeutic agents for stroke are greatly needed.

Owing to the complex and multifaceted nature of the pathophysiology of ischemic stroke, which involves extensive neuronal death and vascular damage along with subsequent inflammatory responses, a number of neuronal cells, glial cells, and endothelial cells are either damaged or lost, leading to disruption of local or global neural networks. Stem cell approach became the most extensively exploited as a treatment for acute or chronic stroke. This is because stem cells and their differentiated cells can either replace lost or damaged neural cells or create a friendly environment for neuroregeneration by mobilizing endogenous neural stem cells upon transplantation. These approaches have shown some promising results in the preclinical and clinical studies for stroke [6,7,8,9,10]. The potential source of therapeutic stem cells includes mesenchymal stromal/stem cells (MSCs, also known as multipotent stromal cells) and neural stem/precursor cells (NPCs), either autologous or allogeneic. While MSCs consistently demonstrated their therapeutic benefits in a number of preclinical studies of neurological disorders [11,12] and thereby supporting their clinical utility, there is no definitive evidence that MSCs differentiated into neural cells in vivo [13,14] and, in many cases, their therapeutic effects may have temporary and confined effects instead of long-lasting improvement in neurological disease [15]. In this regard, the pleiotropic nature of uncommitted neural stem/progenitor cells (NPCs) makes them an ideal source for restoration of this damage in ischemic stroke [16]. Studies have shown that human neural stem cells administered into ischemic rat brains survived, migrated to the ischemic lesion, matured into neurons and glial cells, and restored impaired sensorimotor and spatial learning functions [17,18,19,20].

The minimal requirement for NPCs to be a viable option for stroke therapy is ex vivo culture and expansion of functionally competent NPCs in large quantities. These can be achieved by isolating and culture-expanding fetal neural stem cells or generating them in vitro from pluripotent stem cells (human embryonic stem cells and induced pluripotent stem cells) or somatic cells by direct conversion. Although these NPCs, generated in vitro, may differ from fetal brain-derived NPCs in some aspects, they have been shown to possess therapeutic potential in animal models of neurodegenerative disorders and proven to have therapeutic effects. Consistent with these findings, we reported that transplantation of neural precursor cells (NPCs) derived from human embryonic stem cells (hESCs) significantly promoted behavioral recovery in the ischemic brain of a rat stroke model and induced neuronal cell differentiation of transplanted cells along with a reduction in glial activation and enhancement of angiogenesis in the brain transplanted with NPCs [15]. In the same study, the initial protective and regenerative effects of NPC transplantation appeared to be partly due to paracrine factors from transplanted cells. Since the early phase of NPC-mediated recovery in stroke animal models occurs via paracrine effects, which are mediated by secretory factors, i.e., a collection of cytokines, growth factors, chemokines, angiogenic factors, and extracellular vesicles secreted by stem cells, the utilization of conditioned media (CMs) from NPCs offers several benefits over stem cell-based therapies, including minimal or no adverse effects associated with cell transplantation, such as embolism, ectopic cell proliferation, abnormal tissue formation, tumor formation, immune rejection, and transmission of infectious agents [21,22]. Indeed, CMs collected from stem cells have shown potential to serve as cell-free therapeutic agents in a variety of clinical applications, including various neurodegenerative diseases [23,24,25,26,27,28]

In the present study, we investigated the therapeutic effects of the conditioned medium from PSA-NCAM+NPC (NPC-CM) on the functional recovery and reduction in cerebral infarction after permanent middle cerebral artery occlusion (pMCAO) in rats. In addition, we analyzed the proteomic profile of NPC-CM together with the gene expression profile of brain tissue upon NPC-CM treatment to identify the functional interactome for the tissue protection- and regeneration-associated proteins in the NPC-CM, thereby delineating the underlying therapeutic mechanism of NPC-CM.

## 2. Results

### 2.1. NPC-CM Significantly Reduced the Infarct Area in the Rat Stroke Model

In order to evaluate whether NPC-CM exerts neuroprotective effects on ischemic stroke in rats, the size of the cerebral lesion (infarct volume) was measured on the 15th day after pMCAO (Figure 1a). The infarct volumes caused by pMCAO based on the quantification of metabolically active areas by 2,3,5-triphenyltetrazolium chloride (TTC) versus dead tissue (colorless) of PBS control rats and vehicle (basal media with FGF-2)-treated rats (vehicle control) were 56.3% ± 8.2% and 54.2% ± 10.4%, respectively. Both single NPC-CM treatment (NPC-CM(S)) and multiple NPC-CM treatment (NPC-CM(M)) at 24 h after the onset of ischemic injury significantly decreased brain infarction (29.0% ± 2.0%, *p* < 0.044 for NPC-CM(S) and 24.2% ± 4.1% for NPC-CM(M), respectively, in comparison with the untreated control). The vehicle control showed no therapeutic benefits (Figure 1b). One-way ANOVA demonstrated that rats treated with NPC-CM(S) showed a significantly reduced ischemic area (*p* < 0.05) on day 15 following pMCAO in comparison with the PBS control group, and the rats treated with NPC-CM(M) exhibited significant differences in comparison with the PBS control as well as vehicle-treated control (*p* < 0.007), implying that NPC-CM could protect brain tissue against stroke-induced ischemia (Figure 1c). Although the NPC-CM(M) group appeared to show slightly better infarct size reduction than the NPC-CM(S) group, the difference was not statistically significant.

### 2.2. NPC-CM Improved Behavioral Recovery following pMCAO

Next, we determined whether the observed NPC-CM-mediated reduction in brain infarction contributed to the behavioral recovery in the pMCAO rats. The behavioral tests included a prehensile–traction test, beam balance test, torso twisting test, and rearing test. Weight loss was attenuated by both NPC-CM treatment regimens (NPC-CM(S) and NPC-CM(M)) in comparison with the weight loss in the control rats (PBS and vehicle controls) at 4 and 8 days (Figure 2a). In the prehensile traction test, pMCAO significantly reduced the grasping performance (Figure 2b). However, in comparison with the PBS- and vehicle-treated groups, NPC-CM(S) administration improved the grasping power at 4 and 8 days post-pMCAO, while NPC-CM(M) administration especially increased the endurance on the rope up to 15 days post-pMCAO, implying that NPC-CM administration improved the post-stroke grasping ability of the animals. Although the vehicle controls also showed improvement in the prehensile traction test in comparison with the PBS control, this improvement was not observed from day 8, implying that nutritional and antioxidant components in the vehicle temporarily improved the behavioral function in the acute phase.

NPC-CM(M) also significantly improved beam balance in comparison with the PBS control and vehicle control groups throughout the experimental period (day 3, *p* < 0.001; day 8, *p* < 0.01; day 11, *p* < 0.01; and day 15, *p* < 0.1) (Figure 2c). Additionally, the torso twisting score (Figure. 2d) in the NPC-CM(M) group was significantly better than those in the other groups, implying that the NPC-CM(M) group consistently induced the recovery of paralyzed body parts, which was not observed in NPC-CM(S). The rearing test output (Figure 2e) was also significantly improved in the NPC-CM(M). These data revealed that NPC-CM(M) tended to achieve better scores in the prehensile, beam balance, and torso twisting tests. Collectively, the mNSS (modified neurological severity score) (Figure 2f) showed that NPC-CM(M) induced behavioral recovery for longer durations whereas NPC-CM(S) only yielded a temporary recovery in behavioral scores. Our results indicate that NPC-CM(M) significantly reduced neurological deficit scores by facilitating neural protection, leading to a reduction in stroke lesion area and facilitating functional recovery.

### 2.3. NPC-CM Decreased Inflammation and Induced Neurogenesis in Ischemic Areas

Since ischemic brain damage is mediated by strong inflammatory responses of glial components (astrocytes and microglial cells), we conducted histologic analysis to investigate the effect of NPC-CM on the levels of ED1-expressing activated microglial cells and GFAP-expressing reactive astrocytes in the injured cortex on day 15 after pMCAO (Figure 3a). We observed an accumulation of ED1+-activated microglial cells in the region adjacent to the lesion, and NPC-CM treatment decreased the stroke-induced activation of microglial cells. Quantitatively, the levels of ED1+ microglia were markedly lower in the NPC-CM(M) (60 ± 15.45) and NPC-CM(S)-treated (107 ± 12.32) groups than in the PBS control (185 ± 23.39, *p* < 0.001), Vehicle control(S) (134 ± 15.09), and Vehicle control(M) (136 ± 18.53, *p* < 0.01) groups. Rats treated with NPC-CM(M) had significantly fewer ED1+ cells than those treated with NPC-CM(S) (*p* < 0.1) (Figure 3b). In addition to reduction in ED1+ cell numbers, GFAP+-reactive astrocytes were also significantly reduced in the ischemic striatal region of the NPC-CM(M)- (193 ± 44.27) and NPC-CM(S)-treated (468 ± 136.50) groups in comparison with the PBS control (1354 ± 224.16, *p* < 0.001), Vehicle control(S) (1064 ± 225, *p* < 0.1), and Vehicle control(M) (1223± 185.65, *p* < 0.01), respectively (Figure 3c). Although the level of GFAP+ astrocytes in NPC-CM(M)-treatment was slightly lower than that in the NPC-CM(S)-treatment group, the difference was not statistically significant (*p* > 0.05). CD86-expressing M1 macrophage/microglia is known to be associated with degenerating lesion upon ischemic stroke [29]. Accumulation of M1 microglial cells in the ischemic region was evident in pMCAO rats (Figure 3d). Administration of NPC-CM(S) and NPC-CM(M) greatly reduced the number of cells expressing CD86, a marker for M1 type microglial cells, compared to that of PBS control and vehicle controls (Figure 3e). On the other hand, M2 hallmark Arg1 was significantly increased by NPC-CM(S) and NPC-CM(M). These data suggest that ischemic stroke promotes classical M1 microglia phenotype and NPC-CM significantly attenuated inflammatory response by promoting a beneficial M2 phenotype. Although the number of CD86+ microglial cells tended to decrease in NPC-CM(M) compared to that of NPC-CM(S), the difference did not reach statistical significance.

Next, we sought to determine the effect of NPC-CM on the endogenous neurogenesis in the ischemic region 15 days after pMCAO (Figure 4a,b). For this experiment, brain tissues were stained to identify proliferating and migrating neural progenitor cells (DCX+/BrdU+). The number of DCX+BrdU+ neuroblasts at the level of SVZ was significantly higher in the NPC-CM(S) (44.51 ± 14.2, *p* < 0.01) and NPC-CM(M) (163.3 ± 27.1, *p* < 0.001) in comparison with the PBS control (18.2 ± 3.78), Vehicle control(S) (14.1 ± 3.2), and Vehicle control(M) (163.3 ± 27.1), respectively (Figure 4c). Furthermore, the number of DCX+BrdU+ neuroblasts in the SVZ in NPC-CM(M)-treated rats was significantly greater than that in the NPC-CM(S)-treated animals (*p* < 0.05). These results indicate that NPC-CM(M) augments the endogenous neurogenesis, while this was less significant in NPC-CM(S). Nestin is a well-known marker of neural stem/progenitor cells in the nervous system [30] and increased expression of nestin following ischemic stroke was reported by Nishie et al. [31]. Nestin-positive cells were observed in the SVZ and anterior subventricular zone (aSVZ) of ischemic hemisphere. As of DCX+BrdU+ expressing neuroblast cells in NPC-CM-treated ischemic brains, the number of nestin-positive cells were increased in the NPC-CM(M)-treated brains compared to that of controls and NPC-CM(S) (Figure 4d,e). Taken together, these results showed that NPC-CM(M) strongly induced the mobilization of neural stem/progenitor cells.

### 2.4. NPC-CM Contains Neurotrophic Factors

In order to identify the molecular entities responsible for the therapeutic effects of NPC-CM, we analyzed the protein profiles by mass spectrometry. NPC-CM was collected and probed, as described in Materials and Methods. A total of 760 different proteins from five batches of NPC-CM were identified (Table 1). Of these, 15 proteins were also detected in the vehicle and thus excluded from further analysis. Definition of subcellular localization showed that 67% of the proteins were of the extracellular region (*p* = 1.06 × 10^−157^) including compartments such as the extracellular matrix (*p* = 1.06 × 10^−157^), 24% were from the cytoplasm (*p* = 2.07 × 10^−21^), and 3% were transmembrane proteins (*p* = 9.57 × 10^−40^). Enrichment analysis of protein functions for the extracellular proteins revealed biological processes such as “neural/neuronal”, “regulation of inflammation/immune response”, “extracellular matrix organization”, and “angiogenesis”, which are especially relevant terms for ischemic stroke. The NPC-CM profile contained several key signaling molecules (cytokines and growth factors), such as galectin-1 (Gal1), midkine (MDK), slit homolog 2 (SLIT2), agrin (AGRIN), pleiotrophin (PTN), milk fat globule-EGF factor 8 (MFGE8), and vascular endothelial growth factor beta (VEGFB), which are mostly known to promote angiogenesis and protect cell apoptosis. Some of the secreted proteins included proteases (serpin peptidase inhibitor clade E member 2, SERPINE2; TIMP metallopeptidase inhibitor 2, TIMP2; and matrix metallopeptidase 2, MMP2) are possibly involved in tissue organization and/or cell migration. A number of receptors or membrane proteins were also identified, implying that they are shed to the extracellular space during cell culture.

Next, we performed a predictive protein–protein interaction (PPI) analysis using STRING between the identified proteins in the conditioned medium assuming that some of these interactions may represent key molecules associated with brain tissue repair and regeneration in terms of biological processes (Figure 5a). Of the 760 proteins of NPC-CM, 53 proteins were interconnected and found to be associated with nervous system development, neurogenesis, axon development, regulation of apoptotic process, and angiogenesis. One of the highly detected proteins was midkine, a well-known neurotrophic factor that was also identified in our antibody array (Figure 5b). In addition, the NPC proteome included factors known to be associated with NSC/NPC maintenance, survival, and proliferation, such as PDGF, TGF-β, VEGFB, insulin-like growth factor binding protein 3(IGFBP3) family, plexin-B2 (PLXNB2) and secreted phosphoprotein 1 (SPP1) (data not shown). Collectively, our data indicated the presence of several previously reported components of the NPC-CM [32] and the presence of other potentially important factors produced by NPCs.

### 2.5. NPC-CM Upregulated Transcripts of Anti-Inflammatory Pathway and Neurogenesis

We hypothesized that administration of NPC-CM induces changes in the gene expression profile leading to infarct size reduction and behavioral recovery of stroke animals, especially for genes associated with neural protection, anti-inflammation, angiogenesis, and neurogenesis. In order to delineate the molecular mechanisms underlying the therapeutic effect of NPC-CM, we performed bioinformatics analysis of the differentially regulated genes in the brain tissues at 3 days post-stroke for the PBS control, Vehicle control(S), and NPC-CM(S) groups and 15 days post-stroke for the PBS control, vehicle control(S), Vehicle control(M), NPC-CM(S), and NPC-CM(M) (three animals per group). A comparison of gene expression profiles between the PBS control group and NPC-CM-treated group revealed that a total of 369 genes were differentially expressed in the NPC-CM(S) and NPC-CM(M) groups. When gene expression profiles were compared between the Control and the NPC-CM(S) groups, a total of 185 genes were upregulated in the NPC-CM(S) at 3 days after pMCAO. The most altered biological functions by gene ontology (GO) enrichment analysis were biological regulation (54%), immune system process (16%), and regulation of inflammatory response (5%) (Figure 6a). A combined protein interaction network was constructed with the STRING network within Cytoscape to analyze the interactions of these genes. A total of 185 upregulated genes were highly connected (high confidence = 0.700) within a few clusters, and the top 10 biological processes are shown (Figure 6b). To validate the differentially expressed gene sets from the transcriptome, we selected three prominent genes: two anti-inflammatory cytokines, IL-10 and IL-4, in the GO term of inflammatory responses, and a neurotrophic factor associated with post-stroke recovery and neuronal plasticity, BDNF, in the GO term of nervous system development that are identified as significantly upregulated (>2 fold compared to the PBS control). Their relative expression levels in the rat brain tissue of the PBS control and NPC-CM(S)-treated (of 3 days post-pMCAO) groups were determined in consideration of their possible involvement in the neuroprotective effect [33,34]. On day 3 after pMCAO, the expression levels of IL-10, IL-4, and BDNF were similar in the PBS control and Vehicle control(S). In contrast, the NPC-CM(S) showed significantly higher levels of IL-10, IL-4, and BDNF (Figure 6c). These data confirmed that NPC-CM(S) could modulate the neuroprotective effect in the impaired lesion 3 days post-pMCAO.

A comparison of gene expression profiles between the NPC-CM(S) and the NPC-CM(M) groups showed that a total of 178 genes were upregulated in the NPC-CM(M) at 15 days after pMCAO. GO enrichment analysis revealed several key biological processes, including nervous system development (23%), central nervous system development (13%), neurogenesis (15%), and synaptic transmission (6%) (Figure 7a). A combined protein interaction network generated by STRING protein–protein interaction (PPI) network loaded into Cytoscape showed that 132 of the 178 upregulated genes were highly connected within the network (Figure 7b). The network contained three highly related clusters for neurogenesis, synaptic transmission, and central nervous system development process. The results clearly suggest that the NPC-CM(M) treatment activates part of the neuroregenerative program with molecularly connected modules. Some of the key hubs in the network with multiple connecting genes highlighted transcription factors Tbr2, NeuroD1, Emx1, and Sox5 along with extracellular signaling molecules and neurotrophic factors Bdnf, Reln, Vip, Gal, signal-transducing kinases or nuclear receptors, Nrgn, Nr4a2, and Pgr acting as key hubs between functional modules of the network. Many of these genes are implicated in neurogenesis. Other genes are involved in axon guidance, neurite outgrowth, and neuropeptide signaling/neuronal receptors.

Using real-time RT-PCR of stroke brain tissues, we validated the expression of three selected neuronal genes from the network: Robo3, Lingo1, and NeuroD2, whose expression values were >2 in NPC-CM(M) than in NPC-CM(S) (Figure 7c). Roundabout guidance receptor 3 (Robo3) is known to be associated with neuronal migration [35] and is one of the key downregulated markers for the late-stage animal model of ischemic stroke [36]. Leucine-rich repeat and IgG domain containing protein 1 (Lingo1) is known as a repressor of myelination whose expression is significantly downregulated upon stroke injury due to a loss of precursor and mature oligodendrocytes [37], and its upregulation implies the recovery of oligodendrocyte progenitor and mature oligodendrocytes in the lesions. NeuroD2, a typical marker of neurogenesis and immature neurons [38], was highly expressed by NPC-CM(M), confirming that NPC-CM could modulate the neurogenesis in the impaired lesion 15 days post-pMCAO. BDNF was highly upregulated in NPC-CM(M)-treated brains, indicating that the functional contribution of this neurotrophic factor in the recovering synaptic network in the stroke lesions. Concomitantly, downregulated genes in the NPC-CM-treated rat brains were associated with regulation of inflammatory response, positive regulation of defense response, negative regulation of cytokine production, and extracellular matrix (ECM) organization. The pathways and total gene lists of the downregulated genes discovered are listed in Table 2. With NPC-CM-treatment of ischemic stroke, the most altered biological processes were those involved in inflammation and stroke-induced fibrosis (extracellular matrix organization). Many of the identified genes, including TLR2, TLR3, TLR9, GRN, COL1A1, FN1, and LAMB1, were reported as upregulated genes in previous reports of ischemic stroke [39,40,41,42,43]. As expected, proinflammatory receptors such as IL1RL1 (IL-33), IL1R1, IL17RA, IL18RAP, IL20RB, IL4R, IL6R, and Toll-like receptor families (TLR2, TLR3, TLR4, TLR6, TLR8, and TLR9) were also downregulated by NPC-CM. These changes may represent the reduction in infiltrating inflammatory cells and immune cells to the stroke-damaged sites. In addition, genes contributing to tissue fibrosis (COL1A1, ELN, LOX, TNFRSF1B, LUM, COL4A5, COL4A6, CYP1B1, ADAMTS7, COL18A1, ENG, LCP1, and B4GALT1) were downregulated by NPC-CM. Since excessive production of ECM and basal lamina components (collagens, laminins, and fibronectins) are known to induce tissue fibrosis, robust changes in the expression of these genes at day 15 post-NPC-CM treatment imply the resolution of fibrosis and inflammation. Collectively, NPC-CM treatment led to induction of genes related to biological regulation, system development including neurogenesis, and regulation of system process, while the genes related to immune activation, fibrosis, and loss of cell viability were downregulated during later stages of stoke progression.

## 3. Discussion

In this study, we report that NPC-CM showed significant benefits in the treatment of acute-phase ischemic stroke in rats. Although the regenerative potential of CM or extracellular vesicles derived from different cell types has been described in previous studies [23,28,44,45,46,47,48,49,50], to our knowledge, this is the first study that compared the underlying gene profiles associated with single and multiple injections of NPC-CM in rats with ischemic stroke and assessed the protein profiles of the CM to delineate the therapeutic effects of NPC-CM. The behavioral study showed that NPC-CM improved the neurological outcomes of the pMCAO rat model along with attenuated microglial and astrocyte activation in the ipsilateral side. NPC-CM reduced the number of M1 type microglial cells (CD86+ cell) without affecting M2 phenotype (Arg1+), and this was stronger in the repeated NPC-CM-injected groups. This implies that NPC-CM has an inhibitory effect on proinflammatory microglial cells by shifting their phenotype from M1 to neutral or M2 phenotype via modulating the microenvironment in the lesion. Although blood macrophages and microglial cells share functional and phenotypical characteristics and the blood–brain barrier is not intact in ischemic stroke, the participation of blood macrophages is known to be relatively low compared to that of microglial cells according to previous studies [51,52], implying that M1/M2 detected in the present study is mainly from resident microglial cells. Inhibiting M1 microglial cells while promoting M2 phenotype has been suggested as a potential therapeutic approach for stroke [53]. In addition, the proportion of DCX+/BrdU+ cells in the ipsilateral region of the ischemic brain was significantly higher upon NPC-CM administration in comparison with controls, indicating that NPC-CM induced the mobilization of endogenous neural stem/progenitor cells. In the comparison between single and multiple NPC-CM injections, NPC(M) resulted in substantially more NPCs (DCX+ cells) surrounding the SVZ in the stroke-affected hemisphere. Consistent with the previous finding of increased nestin-positive endogenous neural stem/progenitor cells in SVZ and ischemic region [31], we found that NPC-CM(M) greatly increased the number of nestin+ cells around the SVZ. Thus, our data clearly support the notion that administration of CM derived from therapeutic cells can limit the stroke-induced brain injury by reducing inflammatory responses and enhancing neurogenesis.

The results obtained with the cytokine antibody array demonstrated that the hESC-derived NPC-CM contains an array of neurotrophic factors, such as GDNF, BDNF, and NGF, all of which are known to show neuroprotective effects against pMCAO in rats [54,55]. In order to determine the potential mechanisms underlying the therapeutic effects of NPC-CM, we performed proteome analysis of NPC-CM along with transcriptome analysis of treated rat brains. The results showed that a single i.v. injection significantly reduced infarct volume and resulted in behavioral improvement, and that these events were further enhanced by repeated administration of NPC-CM in a stable and sustained manner in rat ischemic stroke. Apparently, the genes persistently upregulated by both NPC-CM(S) and NPC-CM(M) were involved in central nervous system development (Vsnl1, Nrgn, Vslg, lgsf21), neurogenesis (Sox5, Nrxn1, BDNF, Nrn1, Neurod1, Cck, Robo3), and synapse transmission (Cabp1, Stx1a, Grm6, Rims3), while most of the genes downregulated by NPC-CM treatment were involved in biological processes such as regulation of the inflammatory response, extracellular matrix organization, macrophage activation, and collagen fibril organization. Some of these differentially expressed genes may be molecular targets for stroke therapies. In line with a previous study [36], the time-dependent changes in gene expression profiles in this study were associated with the pathophysiological characteristics of acute and subacute stroke. Downregulation in biological process categories such as inflammatory response, positive regulation of defense response, and ECM organization was observed, while upregulation in biological process categories, including biological regulation (regulation of vascular permeability, negative regulation of acute inflammatory response), system development (neurogenesis, blood vessel development), and regulation of system processes (positive regulation of blood circulation, regulation of tissue remodeling) was evident. Thus, stroke appears to induce the molecular reparative process in the ischemic tissue during days 3 and 14 post-stroke induction. Upon NPC-CM treatment, many of these processes were reversed, and repeated NPC-CM injections further enhanced the tissue-protective and reparative responses to ischemic damage. Cytokine antibody array and proteomic analyses revealed the presence of proteins related to inflammation and immune processes, apoptosis, neural developmental processes, cell adhesion and migration, and responses to stimuli, and many of these proteins may exert their effects through modulation of angiogenesis, neural protection, and neuroregeneration.

NPC-CM(M) and NPC-CM(S) treatment significantly upregulated IL-4, IL-10, and BDNF. Of these, the importance of IL-4 for restoration of neurological function in pathological conditions, including ischemic stroke, has been well documented [56,57]. In addition, IL-4 deficiency is associated with aggravation of brain injury and neurological defects in a stroke animal model [58] indicating its protective role in ischemic stroke. Histologic studies of white matter after stroke injury in IL-4-deficient mice revealed defective remyelination, which was restored by intranasal delivery of IL-4 [59]. Thus, the NPC-CM-dependent upregulation of IL-4 in the ischemic brain in our study appears to play a beneficial role in functional recovery by protecting against neuronal and glial cell death in the white matter. Modulation of post-ischemic inflammation is crucial for recovery from stroke brain injury; thus, interventions against inflammation are an attractive strategy for stroke treatment. IL-10, one of the best known anti-inflammatory cytokines, is known to modulate glial cell activation during pathologic neurodegenerative conditions [60], and to protect neurons and parenchymal cells by suppressing neuroinflammation. In addition, post-ischemic IL-10 gene transfer by adenoviral vector [61] or exogenous IL-10 administration [62] significantly promoted neuroprotection by limiting inflammation in an experimental stroke animal. Thus, the apparent suppression of a devastating inflammatory reaction, as shown by reduced levels of activated astrocytes and M1 microglial cells with increased M2 microglial cell population in the ischemic stroke brain, appears to be mediated by upregulating anti-inflammatory cascades in the ischemic lesion, and IL-10 and IL-4 are clearly the key modulators of the therapeutic effects induced by NPC-CM. Among neurotrophins, the brain-derived neurotrophic factor (BDNF) is known to play a crucial role in the neural development and homeostasis of central nervous system in health and disease. Altered levels of BDNF have been reported in a number of neurodegenerative diseases, including stroke, Parkinson’s disease, and Alzheimer’s disease [57,63,64,65]. In cases involving ischemic stroke, high levels of BDNF were induced during the acute phase while the subsequent decline appears to be strongly associated with clinical outcomes. The diminished level of BDNF led to more severe stroke pathophysiology with reduced angiogenesis [66] and intraventricular injection of BDNF induced neurogenesis [67] in the animal model, implying their roles in neurogenesis and neuronal cell protection during stroke. In our study, NPC-CM(S) induced significant upregulation of these cytokines, and repeated administration further enhanced their level, which was accompanied by functional recovery and stronger neurogenesis in the stroke rat brain. Thus, our findings indicate that these cytokines not only represent potential biomarkers for stroke pathogenesis, but may also serve as novel biological therapeutic agents for neurodegenerative diseases.

Comparative analysis of gene expression profiles between NPC-CM(S)- and NPC-CM(M)-treated rat brains at 15 days after treatment revealed that multiple injections of NPC-CM clearly affected the biological processes underlying nervous system development, central nervous system development, neuron differentiation, neurogenesis, axonogenesis, and synaptic transmission. The tissue expression of four of the genes identified in the gene interactome, Robo3, Lingo1, NeuroD2 and BDNF, was validated by qRT-PCR. The neurogenesis-associated markers Robo3 [35,36], NeuroD2 [38], and BDNF were highly expressed by repeated injection of NPC-CM. A typical marker of the oligodendrocyte population, Lingo1 [37], was significantly upregulated in the NPC-CM(M), confirming that NPC-CM reduced the loss of oligodendrocytes and increased the oligodendrocyte progenitor cell population in the impaired lesion 14 days post-pMCAO. In addition to these four genes acting as hubs of key biological functional clusters in the interaction network, multiple trophic factors were also upregulated in the NPC-CM(M)-treated rats, including Mdk, Bdnf, Reln Vip, Gal, indicating that subacute repair processes are activated to induce neuroreparative responses or neurogenesis upon NPC-CM treatment in the ischemic brain.

## 4. Materials and Methods

### 4.1. Culture of Human hESC-Derived NPC^PSA-NCAM+^

The isolation and subculture of PSA-NCAM-positive NPCs was performed in accordance with the protocol developed by Kim et al. [68]. Briefly, neural differentiation of embryoid bodies (EBs) derived from human embryonic stem cells (SNUhES31, Institute of Reproductive Medicine and Population, Medical Research Center, Seoul National University, Seoul, Korea) was induced by 5 μM dorsomorphin (DM) (Sigma-Aldrich, St. Louis, MO, USA) and 5 μM SB431542 (SB) (Calbiochem, San Diego, CA, USA) in hESC medium deprived of bFGF (Gibco, Grand Island, NY, USA) for 4 days.

PSA-NCAM-positive cells were purified by MACS from expanded neural rosette cells as described previously [69]. Briefly, neural rosette cells treated with 10 μM Y27632 (Sigma-Aldrich) for 1 h were dissociated with Accutase (Invitrogen), and the cells (~1 × 10^8^ cells) were incubated with anti-PSA-NCAM antibody conjugated with microbeads (Miltenyibiotec, Auburn, TX, USA) for 15 min at 4 °C. After washing, the positively labeled cells (NPC^PSA-NCAM+^) were isolated by positive MACS selection and replated on the culture dish at a density of 3.5 × 10^5^ cells/cm^2^ in N2B27 medium or 1× N2, 0.5× B27, and 0.5× G21 supplement (Gemini Bio-Products, West Sacramento, CA, USA) (referred to as NBG medium) plus 20 ng/mL of bFGF.

### 4.2. Preparation of Conditioned Medium of NPC^PSA-NCAM+^(NPC-CM)

After NPC^PSA-NCAM+^ reached 90% confluence in 60 mm dishes containing complete medium, the cells were washed thoroughly three times with 5 mL of PBS and replenished with 5 mL of serum-free DMEM with low glucose supplemented with 1X insulin, transferrin, selenium (ITS, Invitrogen), and 20 ng/mL of bFGF. After culturing for 24 h, NPC-CM was collected, centrifuged at 1000× *g* for 10 min at 4 °C to remove cell debris, and filtered through a membrane with a pore size of 0.22 μm in diameter (Millipore, Billerica, MA, USA). The protein contents were measured by BCA assay, adjusted the concentration to 1 mg/mL and then frozen in aliquots at −80 °C until use.

### 4.3. Permanent Brain Ischemic Stroke Model and NPC-CM Treatment

Adult male Sprague–Dawley rats (body weight, 230–250 g) were obtained from Dae Han Bio Link (Eumseong, Korea). The stroke model was induced by pMCAO as described previously [44]. To ensure consistency in data analysis, we excluded rats showing neither hemiplegia nor neurological deficits at 24 h after the pMCAO procedure. Rats were randomly assigned into one of four groups: PBS control (*n* = 5), vehicle single injection (Vehicle control(S), *n* = 6), vehicle multiple injections (Vehicle control(M), *n* = 4), NPC-CM(S) (*n* = 8), and NPC-CM(M) (*n* = 12). The vehicle control groups received the basal medium containing bFGF. The experimenter responsible for functional evaluation and molecular and histological studies was blinded to the group allocation. Two hundred micrograms of NPC-CM (in 200 μL total) was given to the rats intravenously or basal medium containing bFGF (equal volume of NPC-CM) was administered 24 h after pMCAO. For the multiple-injection group, additional injections of NPC-CM were performed 3, 6, and 11 days after ischemic onset.

### 4.4. Behavioral Tests and Modified Neurological Severity Score (mNSS)

For all animals, a battery of functional tests was performed 24 h before and 24 h after pMCAO and at 1, 4, 8, 11, and 15 days post-NPC-CM injection. All behavioral tests were administered and scored by trained and experienced observers who were blind to the treatment groups of the animals. We assessed the motor weakness of rats subjected to pMCAO with a beam balance test, a torso-twisting test, a prehensile test, a foot fault test, and an open field test [14]. Change in body weight was calculated by subtracting baseline body weight (g) from the body weight measured post-pMCAO. The behavioral evaluation method of pMCAO rat was performed as described previously [44].

### 4.5. Immunohistochemical Analysis and Quantification

After assessing the neurological defect score on day 15, the rats (PBS control, Vehicle control(S), Vehicle control(M), NPC-CM(S), and NPC-CM(M) were anesthetized with zoletil (Virbac SA, Carros, France, 25 mg/kg) and perfused with PBS and 4% paraformaldehyde in PBS (pH 7.5). For infarct volume measurement, the brains were sectioned coronary into 6 slices of 2 mm thickness using a brain matrix slicer. The slices were immediately immersed in a 2% solution of triphenyl-tetrazolium chloride (TTC) (Sigma-Aldrich) for 20 min at 37 °C, followed by fixation in a 4% paraformaldehyde solution overnight before analysis. The stained slices were photographed and analyzed with the NIH Image J program (version 1.47, National Institute of Health, Bethesda, MD, USA). The total infarct volume was calculated using the following equation: the extent of infarction = (infarct volume + ipsilateral undamaged volume contra-lateral volume) × 100/contralateral volume (%) [70].

To study the effects of NPC-CM on the mobilization of endogenous NSCs/NPCs, 5-bromo-20-deoxyuridine (BrdU; 50 mg/kg, Sigma-Aldrich) was administered intraperitoneally twice per day on days 12–14 after pMCAO. For BrdU detection, brain sections were pretreated with 2 N HCl (30 min, 25 °C), neutralized in 0.1 M Tris-buffered saline, and boiled in 10 mM citric acid for 1 h. After preincubation with 0.3% Triton X-100 and 3% normal serum for 30 min, brain sections were incubated with primary antibodies to BrdU (Abcam, Cambridge, UK, at 1:500), doublecortin (DCX; Abcam, at 1:100), ED-1 (Abcam, at 1:100), Nestin (Abcam, at 1:500), CD86 (R&D Systems, at 1:200), Arg-1(Arginase 1, GeneTex, at 1:100), or glial fibrillary acidic protein (GFAP; Millipore, at 1:100). A fluorescent microscope (EVOS M5000; Thermo Fisher Scientific, San Jose, CA, USA) was used to examine the sections and capture fluorescent images. BrdU-, DCX-, GFAP-, and ED-1-positive cells in the SVZ (cells/mm^2^) were quantified using ImageJ software.

### 4.6. Mass Spectrometry-Based Proteome Analysis of NPC-CM

Total protein extraction, two-dimensional electrophoresis, and protein identification were performed at the Yonsei Proteomic Research Center (Seoul, Korea), as described previously [67]. Upon in-gel digestion with trypsin, a nanochip column (Agilent Wilmington, DE, USA, 150 mm × 0.075 mm) was used for peptide separation, and the peptide digests were analyzed by LC-ESI-MS/MS using the EASY-nLC 1000 system (Thermo Fisher Scientific) coupled to LTQ-Orbitrap XL MS (Thermo Fisher Scientific). The chromatography gradient was designed for a linear increase from 0% to 32% in 40 min, 32% to 60% in 4 min, 60% to 95% in 4 min, 95% for 4 min, and 0% in 6 min. The flow rate was maintained at 1500 nL/min. MASCOT (Matrix Science, London, U.K.; version 2.6.0) was used to identify peptide sequences present in the protein sequence database NCBInr (Human). The database search criteria were as follows: (1) MALDI-TOF: NCBInr_Human_20200723 (2) LC-MS/MS: NCBInr_Human_200125. Protein scores higher than 64 for MALDI-TOF or individual ion scores higher than 42 for LC-MS/MS are regarded as significant (*p* < 0.05, FDR < 1%).

### 4.7. Quantitative Real-Time RT-PCR

Total RNA was extracted using TRIzol reagent (Invitrogen), and single-strand cDNA was synthesized from 1 μg of total RNA by using the PrimeScript™ 1st strand cDNA Synthesis Kit (Takara, Tokyo, Japan) in accordance with the manufacturer’s protocol. RT-qPCR was performed using SYBR Green Master Mix (Takara) on a CFX96 (Bio-Rad, Hercules, CA, USA) with the following conditions: 1 min at 95 °C, 15 s at 95 °C, and 20 s at 60 °C. All reactions were measured in triplicate, and the GAPDH level was used as the internal standard. Relative gene expression was determined using the ΔΔCt method. The sequences of the RT-qPCR primers are listed in Table 3.

### 4.8. Total RNA Extraction and RNA-Seq Data Processing from NPC-CM-Treated Rat Brains

RNA was extracted from animals at 3 days post-stroke for the PBS control, Vehicle control(S), and NPC-CM(S) groups and 15 days post-stroke for the PBS control, Vehicle control(S), Vehicle control(M), NPC-CM(S), and NPC-CM(M) from animals processed for RNAseq (3 rats per group). A standard block centered at the territory of the pMCAO (bregma, −1 to +1 mm) was obtained from the ipsilateral region by dissecting from the ipsilateral region on ice. Brain tissue samples were homogenized using the TissueLyser (Qiagen, Temecula, CA, USA.). Total RNA was extracted with TRIZOL Reagent (Invitrogen) in accordance with the manufacturer’s protocol and treated with RNeasy Mini kit (Qiagen). RNA quantity and purity were assessed using the NanoDrop 2000 spectrophotometer (Thermo Fisher scientific), and RNA integrity was assessed using the Agilent 2100 Bioanalyzer (Agilent). All samples had RIN (RNA integrity number) > 8. RNA-seq was performed at Theragen Bio Institute (Suwon, Korea). The cDNA library was prepared with 1~5 μg of total RNA using the Illumina TrueSeq RNA Sample Preparation Kit (Illumina, San Diego, CA, USA) and purified with the QIAQuick PCR purification Kit (Qiagen). Purified cDNA libraries were used for cluster generation and sequenced on the Illumina HiSeq 2500 in accordance with the manufacturer’s protocol. The library was then amplified, and the final library yielded ~400 ng of cDNA, with an average fragment size of ~350 bp. The resulting cDNA libraries (15 samples) were then paired-end sequenced with NextSeq (Illumina).

### 4.9. Functional Analysis of Transcriptomic and Proteomic Data

To determine gene ontology, functional analyses, networks, canonical pathways, and protein–protein interactions of proteins in NPC-CM with stroke recovery-related differentially expressed genes in brain tissues, we performed analyses using the Panther Classification System [71] and the functional analysis and clustering tool from the Database for Annotation, Visualization, and Integrated Discovery (DAVID bioinformatics resources 6.7) [72]. The identified proteins in the NPC-CM were manually filtered for the annotated genes by GO terms related to angiogenesis, anti-inflammation, neurogenesis, and apoptosis, and a list of therapeutic genes was generated. The outcome was connected to these genes by links with Cytoscape-3.8 to generate a therapeutic network, and visualization of the network was performed by STRING (http//string-db.org). Within several biological processes, genes involved in neurogenesis and anti-inflammation were identified for further validation.

### 4.10. Statistical Analysis

All statistical data were analyzed in GraphPad Prism software version 9.0 (GraphPad Software, San Diego, CA, USA) and evaluated using one-way ANOVA. Statistical significance was indicated by *p* < 0.05. One-way ANOVA was corrected by post hoc Fisher’s least significant difference (LSD).

## 5. Conclusions

In conclusion, we showed that multiple injections of NPC-CM significantly ameliorated ischemic stroke injury and neurological deficits by modulating neuroinflammation, supporting neuroprotection, and promoting neuroregeneration by mobilizing endogenous neural stem/progenitor cells. Our proteomic analysis of NPC-CM and transcriptomic analysis of the ischemic brain treated with NPC-CM verified the involvement of previously reported proteins and genes associated with ischemic stroke. Furthermore, our study identified a number of potential molecular targets underlying NPC-CM-mediated therapeutic effects. While NPC-CM (i.e., the NPC secretome) can be a safe and effective alternative to stem cell therapy for stroke and other neurodegenerative diseases, several critical issues still remain to be resolved, including quality control of the produced cells, the development of the clinical culture medium (i.e., the vehicle for the NPC-CM), and the infection/biosafety concerns associated with using this medium for treatment.

## Figures and Tables

**Figure 1 ijms-23-07787-f001:**
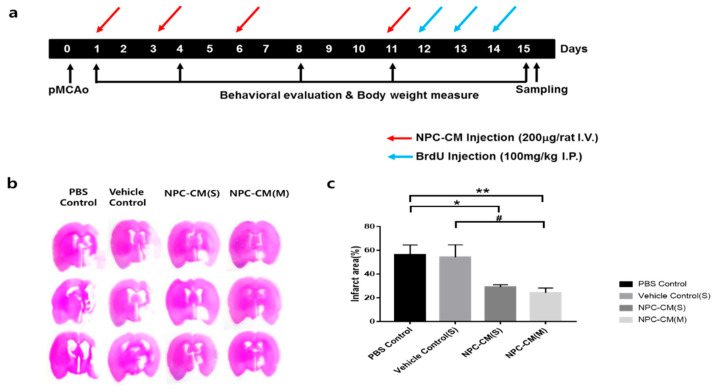
NPC-CM injection reduced the infarct size in a rat stroke model. (**a**) Timeline and experimental design of this study. NPC-CM administration, BrdU injection, and behavioral examination were conducted as indicated. The single injection of NPC-CM(S) was performed one day after pMCAO. Multiple injections of NPC-CM(M) were administered 4 times. Behavioral assessment was performed 5 times after pMCAO. (**b**) Representative 2,3,5-triphenyltetrazolium hydrochloride (TTC)-stained brain sections from rats treated with PBS control, vehicle control (basal medium with FGF-2), NPC-CM(S), and NPC-CM(M) on day 14 after NPC-CM treatment. (**c**) Bar graphs representing the infarct size in NPC-CM-treated stroke models. The infarct volume in the ipsilateral hemisphere is expressed as a percentage of the area of the contralateral hemisphere. Values ± SEM. * *p* = 0.044, ** *p* = 0.007 when compared with the controls. # *p* = 0.033. Control (*n* = 5), vehicle control (*n* = 6), NPC-CM(S) (*n* = 8), NPC-CM(M) (*n* = 8).

**Figure 2 ijms-23-07787-f002:**
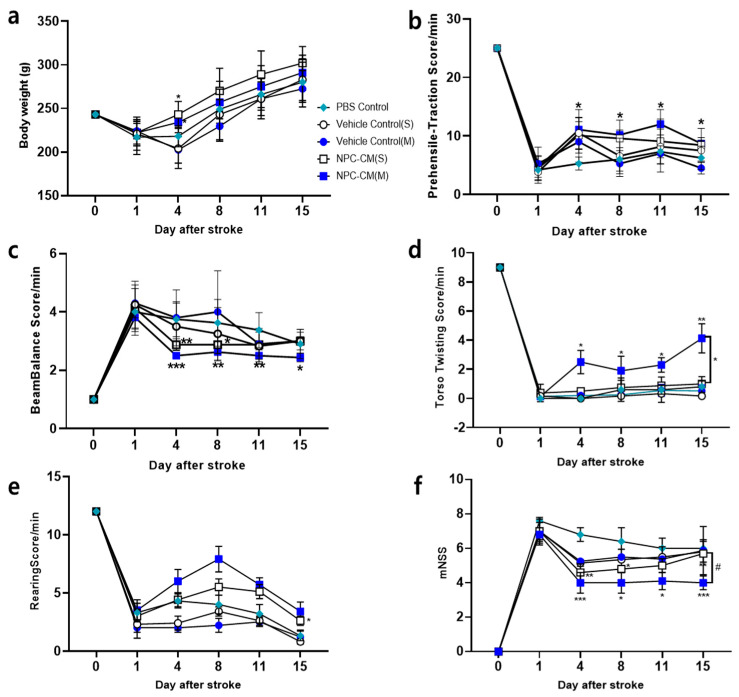
Enhancement of behavioral responses in the rat stroke model after injection of NPC-CM. (**a**) Analysis of body weight changes after treatment with PBS control, Vehicle control(S), Vehicle control(M), NPC-CM(S), and NPC-CM(M). The loss of body weight was higher in the PBS control and vehicle control groups in comparison with those in the NPC-CM(S) and NPC-CM(M)-treated groups. (**b**) Prehensile–traction test. The score was higher in rats injected with NPC-CM(M) at days 4, 8, and 11 (*p* < 0.05). (**c**) Beam balance test. The performance on the beam was scored between 1 and 6. Both NPC-CM groups (NPC-CM(S) and NPC-CM(M)) showed improvements in comparison with the control group. (**d**) Torso twisting test. (**e**) Rearing test in the NPC-CM-treated group was significantly greater than that in the control groups and reached its highest point on day 7. Rearing activity in all groups decreased from day 8 until day 15. (**f**) Modified neurological severity score (mNSS) was determined. Values ± SEM. * *p* ≤ 0.05, ** *p* ≤ 0.01, *** *p* ≤ 0.001, # *p* ≤ 0.05.

**Figure 3 ijms-23-07787-f003:**
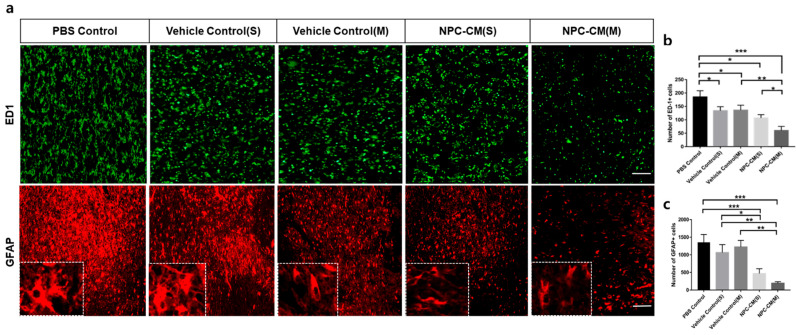
NPC-CM treatment reduced the levels of reactive glia cells at the ischemic lesion in the pMCAO. (**a**) Immunohistochemistry for ED-1 (top) and GFAP (bottom) of the brain from 15 days after pMCAO. Detail of the boxed area showing the astrocyte. (**b**) The expression levels of ED-1 immunostaining in the ischemic brain were reduced in NPC-CM(S) and NPC-CM(M). (**c**) GFAP immunoreactivity in reactive astrocytes at the peri-infarct area of the ipsilateral striatum was significantly decreased in response to NPC-CM(S) and NPC-CM(M) in comparison with those in the PBS control, Vehicle control(S), and Vehicle control(M). (**d**) M1 and M2 microglia in the ischemic lesion was assessed by immunohistochemistry with anti-CD86 and anti-Arg1 antibodies. Counts of CD86+ M1 microglial cells (**e**) and Arg1+ M2 microglial cells (**f**) per mm^2^ in the SVZ ipsilateral to the ischemic injury are shown. Scale bar = 300 μm. Values are indicated as means ± SD. * *p* ≤ 0.05, ** *p* ≤ 0.01, *** *p* ≤ 0.001.

**Figure 4 ijms-23-07787-f004:**
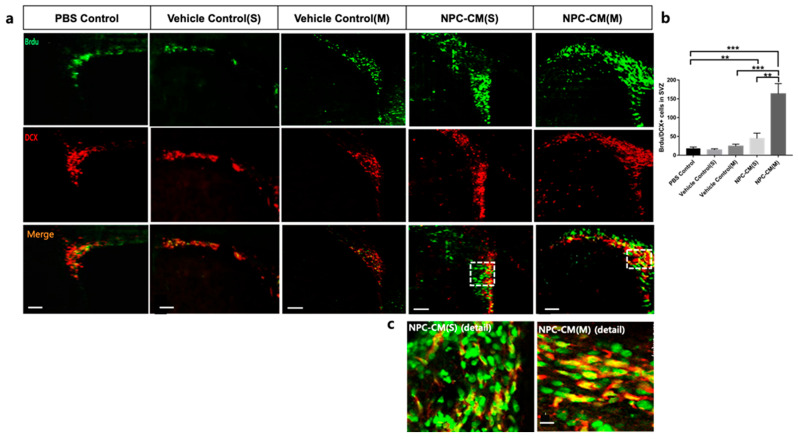
NPC-CM treatment increased endogenous neurogenesis at the ischemic lesion in the pMCAO. (**a**) Immunohistochemistry for BrdU (green) and Doublecortin (DCX) (red) of the rat brain from 15 days after pMCAO. *n* = 4 per group. Scale bar = 300 μm (b) Quantification of BrdU/DCX dual-immunoreactive cells in SVZ from PBS control, vehicle control, NPC-CM(S), and NPC-CM(M). Bar graph revealing the quantification of cells positive for both BrdU and DCX in the SVZ ipsilateral to the ischemic injury. (**c**) Higher magnification of BrdU-labeled cells colocalized with DCX in SVZ (Scale bar = 30 μm). (**d**) Nestin-expressing neural stem/progenitor cells in the SVZ were increased in rats treated with NPC-CM(M). Higher magnification of nestin+ cells in anterior subventricular zone (aSZV) of NPC-CM(M)-treated brain is shown. (**e**) Density of nestin cells per mm^2^ in the SVZ ipsilateral to the ischemic injury is shown. Values are indicated as means ± SD., ** *p* < 0.01, *** *p* < 0.001, *n* = 4 per group. Scale bar = 300 μm.

**Figure 5 ijms-23-07787-f005:**
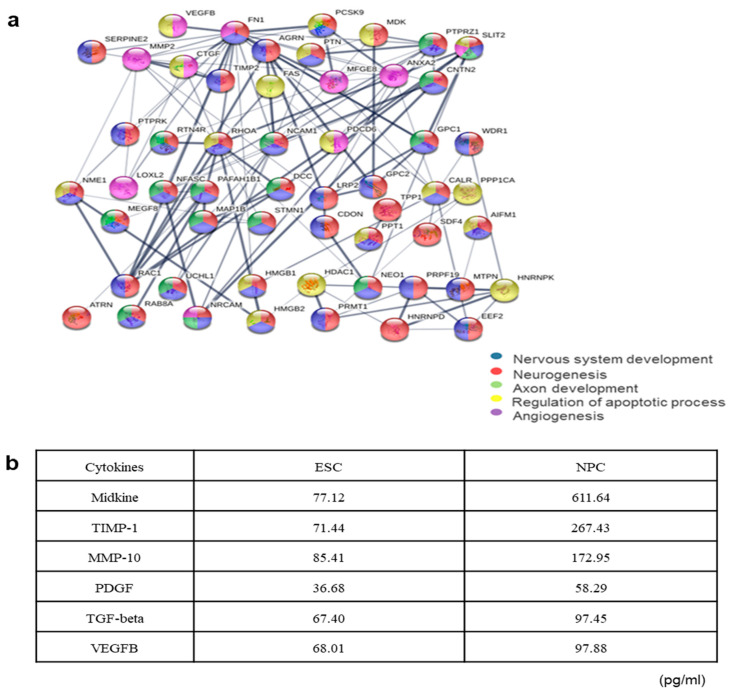
Qualitative analyses of the proteomics findings for NPC-CM. (**a**) Proteome networks showing predicted functional linkages among the identified proteins. STRING analysis showing the protein–protein interactions and biological processes associated with the 53 proteins secreted by NPC. (**b**) Relative levels of key cytokines in hESC-CM and hESC-derived NPC-CM after measurement by a cytokine antibody array (*n* = 3 per group). Conditioned media from hESCs and NPCs were analyzed using a cytokine antibody array, and the relative cytokine levels were measured by quantification of positive pixels for each selected cytokine: midkine; TIMP-1, metallopeptidase inhibitor 1; MMP-10, matrix metalloproteinase-10; PDGF: platelet-derived growth factor; VEGF-B: vascular endothelial growth factor B; TGF-beta: transforming growth factor beta; BDNF: brain-derived neurotrophic factor.

**Figure 6 ijms-23-07787-f006:**
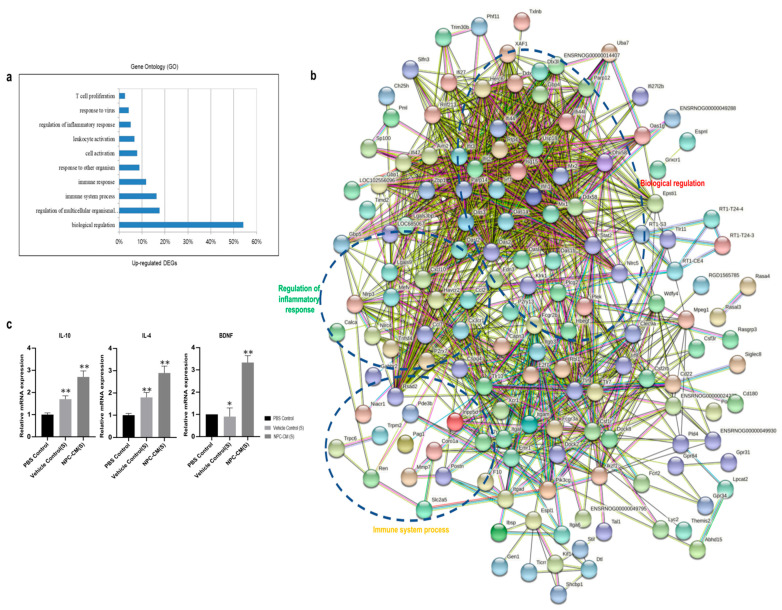
RNAseq transcriptome network of the ischemic brain at day 3 after pMCAO in NPC-CM(S)-treated rats vs. Vehicle control(S)-treated rats. (**a**) Gene ontology enrichment analysis of the NPC-CM-treated group at day 3 after pMCAO. Biological processes that showed the most significant *p*-values in gene ontology analysis are shown. (**b**) Functional network for differentially expressed genes that were associated with the immune system and neuroprotective processes at day 3 after pMCAO. (**c**) Expression levels of upregulated genes in the brain in the NPC-CM(S) assessed by reverse transcription–quantitative polymerase chain reaction; *n* = 3 in each group. * *p* < 0.05, ** *p* < 0.01. IL-10: interleukin-10; IL-4; interleukin-4; BDNF: brain-derived neurotrophic factor.

**Figure 7 ijms-23-07787-f007:**
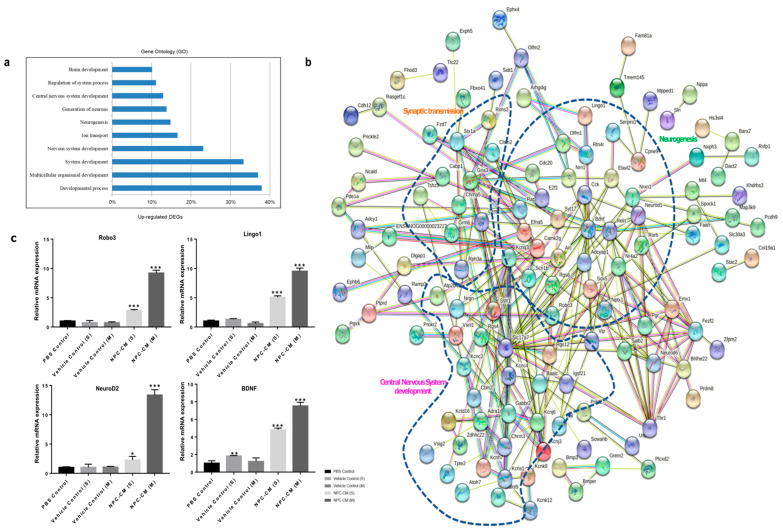
Transcriptome analysis of the ischemic brain at day 15 after pMCAO in NPC-CM(M)-treated rats vs. NPC-CM(S)-treated rats. (**a**) Gene ontology enrichment analysis of the NPC-CM-treated group at day 15 after pMCAO. (**b**) Functional network for differentially expressed genes that were associated with neurological system process at day 15 after pMCAO. (**c**) Expression levels of upregulated genes in the brain of NPC-CM(S) and NPC-CM(M) groups assessed by reverse transcription–quantitative polymerase chain reaction; *n* = 3 in each group. * *p* < 0.05, ***p* < 0.01, *** *p* < 0.001.

**Table 1 ijms-23-07787-t001:** Proteins of the conditioned medium of human PSA-NCAM^+^ neural precursor cells identified by a proteomic approach.

Gene ID	Gene Name	Protein	Family/Subfamily
HUMAN|HGNC = 4852|UniProtKB = Q13547 *	HDAC1	Histone deacetylase 1	HISTONE DEACETYLASE 1 (PTHR10625:SF188)
HUMAN|HGNC = 6972|UniProtKB = P21741	MK	Midkine	MIDKINE (PTHR13850:SF2)
HUMAN|HGNC = 5241|UniProtKB = P11142	HSP7C	Heat shock cognate 71 kDa protein	HEAT SHOCK COGNATE 71 KDA PROTEIN (PTHR19375:SF379)
HUMAN|HGNC = 18540|UniProtKB = Q8TCG5	CATL1	Cathepsin L1	CATHEPSIN L1 (PTHR12411:SF57)
HUMAN|HGNC = 6510|UniProtKB = P16949	STMN1	Stathmin	STATHMIN (PTHR10104:SF5)
HUMAN|HGNC = 5187|UniProtKB = Q99873	ANM1	Protein arginine N-methyltransferase 1	PROTEIN ARGININE N-METHYLTRANSFERASE 1 (PTHR11006:SF54)
HUMAN|HGNC = 12665|UniProtKB = P18206	VINC	Vinculin	VINCULIN (PTHR46180:SF1)
HUMAN|HGNC = 6824|UniProtKB = Q16706	MA2A1	Alpha-mannosidase 2	ALPHA-MANNOSIDASE 2-RELATED (PTHR11607:SF3)
HUMAN|HGNC = 169|UniProtKB = P61160	ARP2	Actin-related protein 2	ACTIN-RELATED PROTEIN 2 (PTHR11937:SF37)
HUMAN|HGNC = 10723|UniProtKB = Q14563	SEM3A	Semaphorin-3A	SEMAPHORIN-3A (PTHR11036:SF23)
HUMAN|HGNC = 26373|UniProtKB = Q2VWP7	PRTG	Protogenin	PROTOGENIN (PTHR12231:SF228)
HUMAN|HGNC = 12692|UniProtKB = P08670	VIME	Vimentin	VIMENTIN (PTHR45652:SF5)
HUMAN|HGNC = 3233|UniProtKB = Q7Z7M0	MEGF8	Multiple epidermal growth factor-like domains protein 8	MULTIPLE EPIDERMAL GROWTH FACTOR-LIKE DOMAINS PROTEIN 8 (PTHR10574:SF287)
HUMAN|HGNC = 8951|UniProtKB = P07093	GDN	Glia-derived nexin	GLIA-DERIVED NEXIN (PTHR11461:SF48)
HUMAN|HGNC = 17104|UniProtKB = Q4KMG0	CDON	Cell adhesion molecule-related/downregulated by oncogenes	CELL ADHESION MOLECULE-RELATED/DOWN-REGULATED BY ONCOGENES (PTHR44170:SF1)
HUMAN|HGNC = 2339|UniProtKB = P29373	RABP2	Cellular retinoic acid-binding protein 2	CELLULAR RETINOIC ACID-BINDING PROTEIN 2 (PTHR11955:SF60)
HUMAN|HGNC = 4449|UniProtKB = P35052	GPC1	Glypican-1	GLYPICAN-1 (PTHR10822:SF8)
HUMAN|HGNC = 12754|UniProtKB = O75083	WDR1	WD repeat-containing protein 1	WD REPEAT-CONTAINING PROTEIN 1 (PTHR19856:SF0)
HUMAN|HGNC = 3594|UniProtKB = P49327	FAS	Tumor necrosis factor receptor superfamily member 6	TUMOR NECROSIS FACTOR RECEPTOR SUPERFAMILY MEMBER 6 (PTHR46874:SF1)
HUMAN|HGNC = 7007|UniProtKB = P61006	RAB8A	Ras-related protein Rab-8A	RAS-RELATED PROTEIN RAB-8A (PTHR47980:SF33)
HUMAN|HGNC = 3720|UniProtKB = Q02790	FKBP4	Peptidyl-prolyl cis-trans isomerase FKBP4	PEPTIDYL-PROLYL CIS-TRANS ISOMERASE FKBP4 (PTHR10516:SF25)
HUMAN|HGNC = 12684|UniProtKB = O15240	VGF	Neurosecretory protein VGF	NEUROSECRETORY PROTEIN VGF (PTHR15159:SF2)
HUMAN|HGNC = 12855|UniProtKB = P63104	1433Z	14-3-3 protein zeta/delta	14-3-3 PROTEIN ZETA/DELTA (PTHR18860:SF7)
HUMAN|HGNC = 7656|UniProtKB = P13591	NCAM1	Neural cell adhesion molecule 1	NEURAL CELL ADHESION MOLECULE 1 (PTHR12231:SF239)
HUMAN|HGNC = 19308|UniProtKB = Q58EX2	SDK2	Sidekick cell adhesion molecule 2	PROTEIN SIDEKICK-2 (PTHR13817:SF59)
HUMAN|HGNC = 1455|UniProtKB = P27797	CALR	Calreticulin	CALRETICULIN (PTHR11073:SF16)
HUMAN|HGNC = 20001|UniProtKB = Q8NBP7	PCSK9	Proprotein convertase subtilisin/kexin type 9	PROPROTEIN CONVERTASE SUBTILISIN/KEXIN TYPE 9 (PTHR43806:SF11)
HUMAN|HGNC = 11086|UniProtKB = O94813	SLIT2	Slit homolog 2 protein	SLIT HOMOLOG 2 PROTEIN (PTHR45836:SF2)
HUMAN|HGNC = 4638|UniProtKB = P09211	GSTP1	Glutathione S-transferase P	GLUTATHIONE S-TRANSFERASE P 1-RELATED (PTHR11571:SF235)
HUMAN|HGNC = 6759|UniProtKB = P29966	MARCS	Myristoylated alanine-rich C-kinase substrate	MYRISTOYLATED ALANINE-RICH C-KINASE SUBSTRATE (PTHR14353:SF9)
HUMAN|HGNC = 9325|UniProtKB = P50897	PPT1	Palmitoyl-protein thioesterase 1	PALMITOYL-PROTEIN THIOESTERASE 1 (PTHR11247:SF8|)
HUMAN|HGNC = 11179|UniProtKB = P00441	SODC	Superoxide dismutase (Cu-Zn)	SUPEROXIDE DISMUTASE [CU-ZN] (PTHR10003:SF66)
HUMAN|HGNC = 5033|UniProtKB = P22626	ROA2	Heterogeneous nuclear ribonucleoproteins A2/B1	HETEROGENEOUS NUCLEAR RIBONUCLEOPROTEINS A2/B1 (PTHR48026:SF13)
HUMAN|HGNC = 30747|UniProtKB = P61201	CSN2	COP9 signalosome complex subunit 2	COP9 SIGNALOSOME COMPLEX SUBUNIT 2 (PTHR10678:SF3)
HUMAN|HGNC = 6871|UniProtKB = P28482	MK01	Mitogen-activated protein kinase 1	MITOGEN-ACTIVATED PROTEIN KINASE 1 (PTHR24055:SF203)
HUMAN|HGNC = 15667|UniProtKB = P58546	MTPN	Myotrophin	MYOTROPHIN (PTHR24189:SF52)
HUMAN|HGNC = 14889|UniProtKB = Q9UBS4	DJB11	DnaJ homolog subfamily B member 11	DNAJ HOMOLOG SUBFAMILY B MEMBER 11 (PTHR44298:SF1)
HUMAN|HGNC = 12852|UniProtKB = P61981	1433G	14-3-3 protein gamma	14-3-3 PROTEIN GAMMA (PTHR18860:SF22)
HUMAN|HGNC = 9674|UniProtKB = Q15262	PTPRK	Protein tyrosine phosphatase receptor type K	RECEPTOR-TYPE TYROSINE-PROTEIN PHOSPHATASE KAPPA (PTHR19134:SF209)
HUMAN|HGNC = 4983|UniProtKB = P09429	HMGB1	High mobility group protein B1	HIGH MOBILITY GROUP PROTEIN B1 (PTHR48112:SF12)
HUMAN|HGNC = 9583|UniProtKB = P26599	PTBP1	Polypyrimidine tract-binding protein 1	POLYPYRIMIDINE TRACT-BINDING PROTEIN 1 (PTHR15592:SF19)
HUMAN|HGNC = 2695|UniProtKB = Q16643	DREB	Drebrin	DREBRIN (PTHR10829:SF1)
HUMAN|HGNC = 6694|UniProtKB = P98164	LRP2	Low-density lipoprotein receptor-related protein 2	LOW-DENSITY LIPOPROTEIN RECEPTOR-RELATED PROTEIN 2 (PTHR22722:SF5)
HUMAN|HGNC = 6836|UniProtKB = P46821	MAP1B	Microtubule-associated protein 1B	MICROTUBULE-ASSOCIATED PROTEIN 1B (PTHR13843:SF5)
HUMAN|HGNC = 7156|UniProtKB = P09238	MMP 10	Stromelysin-2	STROMELYSIN-2 (PTHR10201:SF270)
HUMAN|HGNC = 8574|UniProtKB = P43034	LIS1	Platelet-activating factor acetylhydrolase IB subunit alpha	PLATELET-ACTIVATING FACTOR ACETYLHYDROLASE IB SUBUNIT ALPHA (PTHR44129:SF6)
HUMAN|HGNC = 667|UniProtKB = P61586	RHOA	Transforming protein RhoA	TRANSFORMING PROTEIN RHOA (PTHR24072:SF153)
HUMAN|HGNC = 12412|UniProtKB = Q13885	TBB2A	Tubulin beta-2A chain	TUBULIN BETA-2A CHAIN (PTHR11588:SF100)
HUMAN|HGNC = 2514|UniProtKB = P35222	CTNB1	Catenin beta-1	CATENIN BETA-1 (PTHR45976:SF4)
HUMAN|HGNC = 5258|UniProtKB = P08238	HS90B	Heat shock protein HSP 90-beta	HEAT SHOCK PROTEIN HSP 90-BETA-RELATED (PTHR11528:SF79)
HUMAN|HGNC = 9685|UniProtKB = P23471	PTPRZ	Receptor-type tyrosine-protein phosphatase zeta	RECEPTOR-TYPE TYROSINE-PROTEIN PHOSPHATASE ZETA (PTHR19134:SF461)
HUMAN|HGNC = 5000|UniProtKB = P26583	HMGB2	High mobility group protein B2	HIGH MOBILITY GROUP PROTEIN B2 (PTHR48112:SF3)
HUMAN|HGNC = 2172|UniProtKB = Q02246	CNTN2	Contactin-2	ROUNDABOUT HOMOLOG 2 (PTHR44170:SF9)
HUMAN|HGNC = 17896|UniProtKB = Q9UMS4	PRP19	Pre-mRNA-processing factor 19	PRE-MRNA-PROCESSING FACTOR 19 (PTHR43995:SF1)
HUMAN|HGNC = 29943|UniProtKB = Q9NT68	TEN2	Teneurin-2	TENEURIN-2 (PTHR11219:SF8)
HUMAN|HGNC = 4226|UniProtKB = P31150	GDIA	Rab GDP dissociation inhibitor alpha	RAB GDP DISSOCIATION INHIBITOR ALPHA (PTHR11787:SF3)
HUMAN|HGNC = 4450|UniProtKB = Q8N158	GPC2	Glypican-2	GLYPICAN-2 (PTHR10822:SF24)
HUMAN|HGNC = 329|UniProtKB = O00468	AGRIN	Agrin	AGRIN (PTHR10574:SF375)
HUMAN|HGNC = 8661|UniProtKB = Q9HC56	PCDH9	RCG37051	PROTOCADHERIN-9 (PTHR24028:SF248)
HUMAN|HGNC = 29866|UniProtKB = O94856	NFASC	Neurofascin	NEUROFASCIN (PTHR44170:SF12)
HUMAN|HGNC = 4601|UniProtKB = P28799	GRN	Progranulin	PROGRANULIN (PTHR12274:SF3)
HUMAN|HGNC = 10353|UniProtKB = P36578	RL4	60S ribosomal protein L4	60S RIBOSOMAL PROTEIN L4 (PTHR19431:SF0)
HUMAN|HGNC = 5253|UniProtKB = P07900	HS90A	Heat shock protein HSP 90-alpha	HEAT SHOCK PROTEIN HSP 90-ALPHA-RELATED (PTHR11528:SF87)
HUMAN|HGNC = 132|UniProtKB = P60709	ACTB	Actin, cytoplasmic 1	ACTIN, CYTOPLASMIC 1 (PTHR11937:SF192)
HUMAN|HGNC = 12513|UniProtKB = P09936	UCHL1	Ubiquitin carboxyl-terminal hydrolase isozyme L1	UBIQUITIN CARBOXYL-TERMINAL HYDROLASE ISOZYME L1 (PTHR10589:SF19)
HUMAN|HGNC = 20772|UniProtKB = Q13509	TBB3	Tubulin beta-3 chain	TUBULIN BETA-3 CHAIN (PTHR11588:SF43)
HUMAN|HGNC = 9630|UniProtKB = P21246	PTN	Pleiotrophin	PLEIOTROPHIN (PTHR13850:SF1)
HUMAN|HGNC = 20637|UniProtKB = Q9BPU6	DPYL5	Dihydropyrimidinase-related protein 5	DIHYDROPYRIMIDINASE-RELATED PROTEIN 5 (PTHR11647:SF58)
HUMAN|HGNC = 2701|UniProtKB = P43146	DCC	Netrin receptor DCC	NETRIN RECEPTOR DCC (PTHR44170:SF8)
HUMAN|HGNC = 12851|UniProtKB = P62258	1433E	14-3-3 protein epsilon	14-3-3 PROTEIN EPSILON (PTHR18860:SF17)
HUMAN|HGNC = 9760|UniProtKB = P62491	RB11A	Ras-related protein Rab-11A	RAS-RELATED PROTEIN RAB-11A (PTHR47979:SF49)
HUMAN|HGNC = 7849|UniProtKB = P15531	NDKA	Nucleoside diphosphate kinase A	NUCLEOSIDE DIPHOSPHATE KINASE A (PTHR11349:SF69)
HUMAN|HGNC = 9087|UniProtKB = Q04941	A4	Amyloid-beta A4 protein	AMYLOID-BETA PRECURSOR PROTEIN (PTHR23103:SF7)
HUMAN|HGNC = 3214|UniProtKB = P13639	EF2	Elongation factor 2	ELONGATION FACTOR 2 (PTHR42908:SF27)
HUMAN|HGNC = 652|UniProtKB = P84077	ARF1	ADP-ribosylation factor 1	ADP-RIBOSYLATION FACTOR 1 (PTHR11711:SF357)
HUMAN|HGNC = 10820|UniProtKB = O60880	SAP	Prosaposin	PROSAPOSIN (PTHR11480:SF36)
HUMAN|HGNC = 11820|UniProtKB = P01033	TIMP1	Metalloproteinase inhibitor 1	METALLOPROTEINASE INHIBITOR1(PTHR11844:SF24)
HUMAN|HGNC = 2095|UniProtKB = P10909	CLUS	Clusterin	CLUSTERIN (PTHR10970:SF1)
HUMAN|HGNC = 7756|UniProtKB = P48681	NEST	Nestin	NESTIN (PTHR47051:SF1)
HUMAN|HGNC = 10440|UniProtKB = P62081	RS7	40S ribosomal protein S7	40S RIBOSOMAL PROTEIN S7 (PTHR11278:SF5)
HUMAN|HGNC = 20|UniProtKB = P49588	SYAC	Alanine–tRNA ligase, cytoplasmic	ALANINE--TRNA LIGASE, CYTOPLASMIC (PTHR11777:SF34)
HUMAN|HGNC = 7637|UniProtKB = P55209	NP1L1	Nucleosome assembly protein 1-like 1	NUCLEOSOME ASSEMBLY PROTEIN 1-LIKE 1 (PTHR11875:SF70)
HUMAN|HGNC = 3014|UniProtKB = Q16555	DPYL2	Dihydropyrimidinase-related protein 2	DIHYDROPYRIMIDINASE-RELATED PROTEIN 2 (PTHR11647:SF56)
HUMAN|HGNC = 8768|UniProtKB = O95831	AIFM1	Apoptosis-inducing factor 1, mitochondrial	APOPTOSIS-INDUCING FACTOR 1, MITOCHONDRIAL (PTHR43557:SF4)
HUMAN|HGNC = 2730|UniProtKB = Q08345	DDR1	Epithelial discoidin domain-containing receptor 1	EPITHELIAL DISCOIDIN DOMAIN-CONTAINING RECEPTOR 1 (PTHR24416:SF333)
HUMAN|HGNC = 7994|UniProtKB = Q92823	NRCAM	Neuronal cell adhesion molecule	NEURONAL CELL ADHESION MOLECULE (PTHR10075:SF44)
HUMAN|HGNC = 9670|UniProtKB = P10586	PTPRF	Receptor-type tyrosine-protein phosphatase F	RECEPTOR-TYPE TYROSINE-PROTEIN PHOSPHATASE F (PTHR19134:SF203)
HUMAN|HGNC = 6743|UniProtKB = Q14444	CAPR1	Caprin-1	CAPRIN-1 (PTHR22922:SF3)
HUMAN|HGNC = 1874|UniProtKB = P23528	COF1	Cofilin-1	COFILIN-1 (PTHR11913:SF17)
HUMAN|HGNC = 3778|UniProtKB = P02751	FINC	Fibronectin	FIBRONECTIN (PTHR19143:SF267)
HUMAN|HGNC = 1759|UniProtKB = P19022	CADH2	Cadherin-2	CADHERIN-2 (PTHR24027:SF79)
HUMAN|HGNC = 914|UniProtKB = P61769	B2MG	Beta-2-microglobulin	BETA-2-MICROGLOBULIN (PTHR19944:SF62)
HUMAN|HGNC = 10549|UniProtKB = Q9UBB4	ATX10	Ataxin-10	ATAXIN-10 (PTHR13255:SF0)
HUMAN|HGNC = 1763|UniProtKB = P55283	CADH4	Cadherin-4 (fragment)	CADHERIN-4 (PTHR24027:SF81)
HUMAN|HGNC = 18601|UniProtKB = Q9BZR6	RTN4R	Reticulon-4 receptor	RETICULON-4 RECEPTOR (PTHR45836:SF6)
HUMAN|HGNC = 381|UniProtKB = P15121	ALDR	ATP-binding cassette subfamily D member 2	ATP-BINDING CASSETTE SUB-FAMILY D MEMBER 2 (PTHR11384:SF24)
HUMAN|HGNC = 4458|UniProtKB = P06744	G6PI	Glucose-6-phosphate isomerase	GLUCOSE-6-PHOSPHATE ISOMERASE (PTHR11469:SF1)
HUMAN|HGNC = 10420|UniProtKB = P23396	RS3	40S ribosomal protein S3	40S RIBOSOMAL PROTEIN S3 (PTHR11760:SF32)
HUMAN|HGNC = 1096|UniProtKB = O95861	PIP	Prolactin-inducible protein homolog	PROLACTIN-INDUCIBLE PROTEIN (PTHR15096:SF5)
HUMAN|HGNC = 11655|UniProtKB = P17987	TCPA	T-complex protein 1 subunit alpha	T-COMPLEX PROTEIN 1 SUBUNIT ALPHA (PTHR11353:SF84)
HUMAN|HGNC = 9281|UniProtKB = P62136	PP1A	Serine/threonine-protein phosphatase PP1-alpha catalytic subunit	SERINE/THREONINE-PROTEIN PHOSPHATASE PP1-ALPHA CATALYTIC SUBUNIT (PTHR11668:SF377)
HUMAN|HGNC = 12729|UniProtKB = P23381	SYWC	Tryptophan--tRNA ligase, cytoplasmic	TRYPTOPHAN--TRNA LIGASE, CYTOPLASMIC (PTHR10055:SF1)
HUMAN|HGNC = 6666|UniProtKB = Q9Y4K0	LOXL2	Lysyl oxidase homolog 2	LYSYL OXIDASE HOMOLOG 2 (PTHR45817:SF1)
HUMAN|HGNC = 8765|UniProtKB = O75340	PDCD6	Programmed cell death protein 6	PROGRAMMED CELL DEATH PROTEIN 6 (PTHR23064:SF41)
HUMAN|HGNC = 537|UniProtKB = P07355	ANXA2	Annexin A2	ANNEXIN A2-RELATED (PTHR10502:SF18)
HUMAN|HGNC = 7036|UniProtKB = Q08431	MFGM	Lactadherin	LACTADHERIN (PTHR24543:SF291)
HUMAN|HGNC = 13518|UniProtKB = Q9Y696	CLIC4	Chloride intracellular channel protein 4	CHLORIDE INTRACELLULAR CHANNEL PROTEIN 4 (PTHR43920:SF7)

*** HGNC: Human Gene Nomenclature Committee.

**Table 2 ijms-23-07787-t002:** GO enrichment analysis and biological processes for the downregulated genes in NPC-CM(M)-treated ischemic rat brains.

**Biological Process**	**Gene Ontology ID**	**Adjusted *p*-Value**	Gene Symbol
Regulation of inflammatory response	GO:0050727	1.46 × 10^−^^28^	CTSC,ANXA1,SERPINF1,SERPINE1,FCER1G,ADAMTS12,LBP,T NFRSF1B,ACP5,IL20RB,TREM2,IGF1,ALOX5AP,FCGR3,FABP4,P YCARD,NLRP3,FCGR1A,CEBPA,PIK3AP1,SIGLEC10,GPX1,LYN, FCGR2,STAP1,TLR2,IL1RL1,LACC1,TMEM173,CNR2,FUT7,TNFR SF1A,TLR3,ZFP36,TLR6,HGF,VAMP8,GRN,AOAH,TRADD,BTK,I L1R1,IL17RA,ENPP3,CASP1,PLA2G2D,METRNL,CDH5,C1QTNF 3,TLR9,ETS1,GPR31,IFI35,TLR4,USP18,NT5E,TGM2,PTGES,NFKB IA,NFKBIZ,PTGER4,TNFSF4,SLAMF8,ADA,SBNO2,ATM,CD276 ,SOCS3,DAGLB,NFKB1,CALCRL,BST1,RIPK1
Positive regulation of defense response	GO:0031349	1.70 × 10^−^^16^	CTSC,SERPINE1,FCER1G,SPI1,LBP,VAV1,TREM2,TLR8,ALOX5A P,FCGR3,TYROBP,FABP4,PYCARD,FCGR1ACEBPA,STAP1,TLR2, IL1RL1,TMEM173,TNFRSF1A,TLR3,HAVCR2,TLR6,RGD1565785, VAMP8,TRADD,BTKFCNB,IL17RA,IL18RAP,TLR9,ETS1,IFI35,TL R4,MNDA,ZBP1,TGM2,NFKBIA,IRGM,NFKBIZ,PTGER4,GBP5,P ARP9,NOD1,PLSCR2,NLRC5,ARG1,RIPK1,CYBA
Negative regulation of cytokine production	GO:0001818	4.97 × 10^−^^16^	LILRB4,ANXA1,SERPINB1A,LBP,PTPN6,ACP5,IL20RB,TREM2,T LR8,CD84,IGF1,FURIN,PYCARD,TGFB1,NLRP3,LRRC32,FCGR2, NCKAP1L,TLR2,IL1RL1,CMKLR1,C5AR2,LAPTM5,HAVCR2,ZFP 36,TLR6, FN1,HGF,BTK,GPNMB,C1QTNF3,TLR9,TLR4,DHX58,APOD,BCL 3,RELB,PTGER4,TNFSF4,ZC3H12A,CSK,CD276,MERTK,ARRB2,I NPP5D,AXL,ABCD1,NFKB1,ARG1,CD33,ANGPT1
Extracellular matrix organization	GO:0030198	8.72 × 10^−^^16^	COL1A1,ELN,LOX,TNFRSF1B,LUM,LOXL1,COL4A5,COL4A6,CY P1B1,ADAMTS7,COL18A1,ANXA2,NID1,ENG,LCP1,B4GALT1,T GFB1,FAP,FOXC1,COL15A1,FKBP10,AEBP1,ADAMTS15,TNFRSF 1A,FBLN1,COLGALT1,CCDC80,OLFML2B,FMOD,COL5A2,COL3 A1,SERPINH1,SLC39A8,TIE1,LAMC1,ADAMTS14,CTSS,ITGB1,L OC102555086,MYO1E,KAZALD1,COL6A1,DDR2,TGFBI,CREB3L1 ,FBLN5,POSTN,OLFML2A,PDGFRA,LAMB2,PRDX4,EMILIN1,C OL11A1,ADAMTS2
Macrophage activation	GO:0042116	1.45 × 10^−^^12^	CTSC,C5AR1,AIF1,LBP,C1QA,TREM2,CD84,SYK,CEBPA,STAP1,T LR2,IL1RL1,IL4R,TLR3,TMEM106A,HAVCR2,TLR6,ITGAM,IFNG R1,PLA2G4A,TLR9,IFI35,TLR4,IFNGR2,TLR1,SBNO2,ATM
T cell differentiation	GO:0030217	1.70 × 10^−^^9^	LILRB4,ANXA1,FCER1G,SPI1,VAV1,HLX,MAFB,SASH3,SYK,IKZ F1,TGFB1,NLRP3,MPZL2,CD8A,NCKAP1L,CYP26B1,IL4R,FUT7, TCIRG1,CORO1A,WNT4,PLA2G2D,IL6R,ZFP36L1,CTSL,FGL2,IL 2RG,CBFB,RHOH,TGFBR2,RELB,NFKBIZ,PTGER4,LFNG,TNFSF4 ,CD4,ZC3H12A,ADA,IRF1,PRDM1,FZD7,B2M,CDK6
Collagen fibril organization	GO:0030199	2.32 × 10^−^^9^	COL1A1,LOX,LUM,LOXL1,CYP1B1,ANXA2,FOXC1,FKBP10,AEB P1,COLGALT1,FMOD,COL5A2,COL3A1,SERPINH1,ADAMTS14, LOC102555086,DDR2,EMILIN1,COL11A1,ADAMTS2
Aging	GO:007568	0.002424	CTSC,CD68,SERPINF1,SERPINE1,SPI1,C1QA,ITGB2,CD86,CDK1, CCL2,CFH,GJB2,CTSL,BAK1,APOD,PRKCD,CDKN1C,RBL1,CDK N1A,AURKB,ADM,ADA,NFE2L2,FBXO5,ATM,IFI27L2B,INPP5D, BCL2A1,B2M,ARG1,CCL11,CDK6
Negative regulation of blood vessel morphogens	GO:2000181	0.0022754	THBS2,SERPINF1,SERPINE1,DCN,FOXC1,CCL2,WNT4,CXCL10, HHEX,STAB1,TIE1,ANGPT4,CREB3L1,FBLN5,EMILIN1,PDE3B

**Table 3 ijms-23-07787-t003:** Genes and primer sequences used for qRT-PCR.

Gene	Primer Sequence
IL-4	Forward: 5′-TGCACCGAGATGTTTGTACC-3′
	Reverse: 5′-GGATGCTTTTTAGGCTTTCC-3′
IL-10	Forward: 5′-GCAGGACTTTAAGGGTTACTTGG-3′
	Reverse: 5′-GGGGAGAAATCGATGACAGC-3′
BDNF	Forward: 5′-TGGGGTTAGGAGAAGTCAAGC-3′
	Reverse: 5′-TGTTTCACCCTTTCCACTCCT-3′
ROBO3	Forward: 5′-ACCCTGATGCTGCACTTCTGG-3′
	Reverse: 5′-TCCGGCTTCGGCTGCGT-3′
LINGO1	Forward: 5′-AGAGACATGCGATTGGTGA-3′
	Reverse: 5′-AGAGATGTAGACGAGGTCATT-3′
NEUROD2	Forward: 5′-CAAGAAGCGCGGGCCGAAGA -3′
	Reverse: 5′-TTGGCCTTCTGTCGCCGCAG -3′

## Data Availability

Not applicable.

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
