# Peer review of "Conditioned Medium of Human Pluripotent Stem Cell-Derived Neural Precursor Cells Exerts Neurorestorative Effects against Ischemic Stroke Model"

_ijms, 2022, doi:10.3390/ijms23147787_

Round 1

Reviewer 1 Report

In the present study, authors have sought to investigate the roles of NPC-CM in rates following ischemic stroke. Authors showed that administration of NPC-CM significantly improved neurological functions with reduced injured areas. In addition, authors showed that NPC-CM injection decreased inflammation and gliosis, while it promoted neurogenesis. Furthermore, proteome and/or GO analysis showed that several factors (e.g., related to neurogenesis, angiogenesis, nervous system development) were enriched in NPC-CM. In general, this paper is interesting and well organized. I made only a few comments to improve the quality of this manuscript.

1. Authors showed that NPC-CM injection reduced the numbers of ED1+ microglia compared with controls. Did NPC-CM regulate M1/M2 polarization of microglia? Did NPC-CM reduce the numbers of M1 microglia or increase the numbers of M2 microglia?

2. Authors showed that NPC-CM injection increased the numbers of DCX+/BrdU+ cells assessed at the SVZ compared with controls. A recent study using Nestin-GFP transgenic mice shows that endogenous neural stem/progenitor-like cells were regionally induced within and around the ischemic areas as well as the SVZ (Nishie et al, International Journal of Molecular Sciences, 22, 12997, 2021). Did NPC-CM injection increase the numbers of nestin+ cells at these regions?

3. Line 121, 123: Please correct “NPC(S)” into “NPC-CM(S)” if it is misspelling.

Reviewer 2 Report

In an effort to explore the therapeutic potential of NPC-CM in the treatment of acute ischemic stroke, the authors conducted a series of conditioned medium injections in pMCAO rats. They evaluate behavioral response to treatment, infarct size, levels of astrogliosis and neurogenesis, and the immunomodulatory effect of CM treatment.

The manuscript is based on a significant amount of experimental (mostly in vivo) data, which makes this study of great value. However, there are several major issues concerning the experimental design.

Major:

1.       In the "Method" section, the dose of CM administration is indicated as 0.2 mg/kg of rat body weight. When converting mass to volume, a simple arithmetic calculation can be obtained that one injection was about 0.04 μl per animal. Considering the significant positive effect of intravenous CM injections on behavioral response, infarct volume (almost 2-fold reduction) and other parameters, it seems incredible that such a small amount of CM can be so useful. I believe this may be a mistake (CM could have been concentrated, or the concentration could have been expressed as total protein per unit volume), but the authors must clearly describe the method of preparation and dosage of CM, otherwise the overall design of the study is suspect.

Minor:

2.       Figure 3a and d high magnification images should be provided to demonstrate the specific morphology of the named cells (astrocytes and neuroblasts).

3.       In conclusion, the authors state that "multiple injections of NPC-CM significantly improve the condition after ischemic stroke ... due to ... restoration of vascular networks." Histological evidence of vasculature repair must be presented or this statement should be excluded from the conclusion.

Round 2

Reviewer 1 Report

I agree with authors' comment. I think authors significantly improved their manuscript. 

Reviewer 2 Report

I agree, I think the authors have greatly improved the manuscript